# CORGI[2]: A HYBRID OFFLINE-ONLINE APPROACH TO STORAGE-AWARE DATA SHUFFLING FOR SGD

## ABSTRACT

When using Stochastic Gradient Descent (SGD) for training machine learning models, it is often crucial to provide the model with examples sampled at random from the dataset. However, for large datasets stored in the cloud, random access to individual examples is often costly and inefficient. A recent work (Xu et al., 2022a), proposed an online shuffling algorithm called CorgiPile, which greatly improves efficiency of data access, at the cost some performance loss, which is particularly apparent for large datasets stored in homogeneous shards (e.g., video datasets). In this paper, we introduce a novel two-step partial data shuffling strategy for SGD which combines an offline iteration of the CorgiPile method with a subsequent online iteration. Our approach enjoys the best of both worlds: it performs similarly to SGD with random access (even for homogenous data) without compromising the data access efficiency of CorgiPile. We provide a comprehensive theoretical analysis of the convergence properties of our method and demonstrate its practical advantages through experimental results. paragraph.

## 1 INTRODUCTION

Modern machine learning pipelines, used for training large neural network models, require extensive datasets which are frequently stored on cloud-based systems due to their size surpassing the capacity of fast memory. Stochastic gradient descent (SGD) and its variants have emerged as the primary optimization tools for these models. However, SGD relies on independent and identically distributed (i.i.d.) access to the dataset, which is advantageous when random access memory is available. In contrast, when utilizing slower storage systems, particularly cloud-based ones, random access becomes costly, and it is preferable to read and write data sequentially (Aizman et al., 2019).

This challenge is further compounded by the customary storage of data in shards (Bagui & Nguyen, 2015), which are horizontal (row-wise) partitions of the data. Each partition is maintained on a separate server or storage system to efficiently distribute the load. For example, image data is often acquired in the form of videos, leading to the storage of single or multiple clips within each shard. This results in highly homogeneous or non-diverse chunks of data. Consequently, running SGD with sequential reading of examples, without randomized access, will result in a suboptimal solution (Xu et al., 2022a; Nguyen et al., 2022).

One could shuffle the dataset fully before performing SGD, but this also requires random access to memory. A recent line of work (Feng et al., 2012; Google, 2022; Nguyen et al., 2022) suggests an attractive alternative: performing a partial online shuffle during training time. In particular, (Xu et al., 2022a) recently suggested the CorgiPile shuffling algorithm, a technique that reads multiple shards into a large memory buffer, shuffles the buffer, and uses the partially shuffled examples for training. This approach gains data access efficiency, albeit at the expense of performance loss that is especially noticeable for large datasets stored in homogeneous shards. The goal of this work, is to find a better trade-off between data access efficiency and optimization performance.

In this paper, we present Corgi[2], a novel approach that enjoys the strengths of both offline and online data shuffling techniques. We propose adding another offline step that incurs a small overhead compared to Corgi (much cheaper than a full offline shuffle). Thus, our method entails a two-step partial data shuffling strategy, wherein an offline iteration of the CorgiPile method is succeeded by a subsequent online iteration (see schematic representation in Figure 2). This approach enables us to

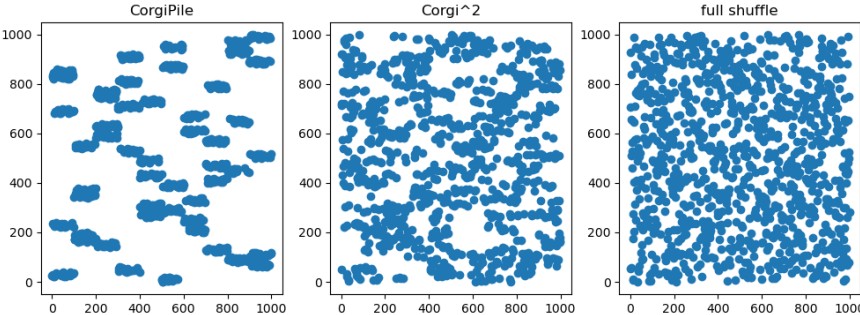

Figure 1: The results of a simulation illustrate a a qualitative comparison of shuffling the set $\{1, ..., 1000\}$ using CorgiPile, $\text{Corgi}^2$ and full shuffle. It is intuitively clear that $\text{Corgi}^2$ is close to a uniform shuffling of the data.

achieve performance comparable to SGD with random access, even for homogeneous data, without compromising the data access efficiency of CorgiPile. In Figure 1 we illustrate the distribution of data indices after CorgiPile, $\text{Corgi}^2$ and Full Shuffle. It is intuitively clear that $\text{Corgi}^2$ spreads the data samples more uniformly across the shuffled buffer.

We perform a comprehensive theoretical analysis of the convergence properties of our method, demonstrating its compatibility with SGD optimization. We further underscore the practical advantages of our approach through a series of experimental results, highlighting its potential to improve the way we train machine learning models in storage-aware systems.

**Our Contributions.** We delineate our contributions as follows:

1. In Section 2, we introduce $\text{Corgi}^2$, a novel two-step partial data shuffling strategy that combines the strengths of both offline and online data shuffling techniques.

2. We provide a comprehensive theoretical analysis of the convergence properties of our method in Section 3, demonstrating its robustness and efficacy. Our analysis elucidates the conditions under which $\text{Corgi}^2$ converges and offers insights into the trade-offs between data access efficiency and optimization performance.

3. We conduct a series of experiments to empirically validate the effectiveness of $\text{Corgi}^2$ in Section 4. Our results underscore the practical advantages of $\text{Corgi}^2$ in various settings.

**Related Work.** It is well established that SGD works well for large datasets given i.i.d access to the data (Bottou, 2010; Brown et al., 2020; He et al., 2021). Since such access is costly in practice, it is often simulated by fully shuffling the dataset "offline" (before training), and reading the data sequentially "online" (during training). While this requires a lengthy and expensive offline phase, prior work demonstrated that the resulting convergence rate of training is comparable to that of random access SGD (Shamir, 2016; Safran & Shamir, 2020; Gurbuzbalaban et al., 2019; 2021).

Recently, many approaches for "partial" shuffling have emerged, attempting to compromise on the uniformity of randomness in exchange for reduced sampling costs. Prominently, Tensorflow implements a sliding-window shuffling scheme that reads the data sequentially into a shuffled buffer. Other approaches include Multiplexed Reservoir Sampling (MRS) (Feng et al., 2012), where two concurrent threads are used to perform reservoir sampling and update a shared model and Nguyen et al. (2022), where multiple workers read chunks of data in parallel and communicate shuffled data between each other. Finally, Xu et al. (2022a) recently introduced the highly effective CorgiPile algorithm, which compares favorably to all the other approaches we're aware of, and which we seek to directly improve upon by prepending an *efficient, partial* offline shuffle step to it.

While our attention focuses on the impact of $\text{Corgi}^2$ on the machine learning training process, Nestoridi & White (2016) analyzes the mixing times of a similar shuffle method for a deck of cards.

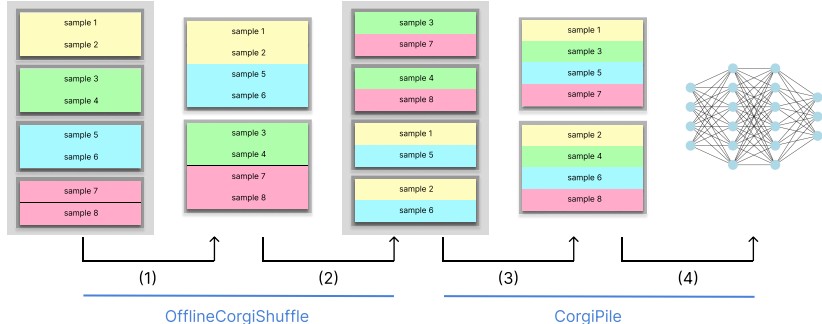

Figure 2: Schematic visualization of the end-to-end flow of Corgi[2]. Samples of the same color belong to the same storage block before the process starts. (1) Sets of blocks are randomly selected from the datasets and stored in a local buffer. (2) The buffer is randomly shuffled and written in new blocks. (3) During training, sets of the new blocks are loaded at random into a buffer. (4) Each buffer is shuffled and processed with SGD.

## 2 SETTING AND ALGORITHM

In this section, we present our notation and framework, which encapsulates the formalization of diverse machine learning tasks. Consider the following standard optimization problem, where our goal is to minimize an average of functions $\{f_1, \ldots, f_m\}$:

$$\min_{\mathbf{x}} F(\mathbf{x}) = \frac{1}{m} \sum_{i=1}^{m} f_i(\mathbf{x}) \tag{1}$$

For instance, consider a dataset of $m$ examples, and let $f_i$ represent the loss over the $i$-th example. Then, the objective function $F(\mathbf{x})$ is the average loss over the individual examples across the entire dataset. Our task is to find the optimal value of $\mathbf{x}$ (e.g., the parameters of some machine learning model) that minimizes this objective function.

Many of the prevalent modern approaches to machine learning involve optimizing this objective function using the Stochastic Gradient Descent (SGD) algorithm. An execution of SGD initializes the parameter vector $\mathbf{x}_0$ and then performs $\mathcal{T}$ *epochs*, each consisting of multiple iterations of the following procedure:

1. Sample an example $f_i$ from the dataset, where $i$ is selected uniformly at random.

2. Compute the gradient $\nabla f_i(\mathbf{x}_{j-1})$ and update $\mathbf{x}_j = \mathbf{x}_{j-1} - \eta_j \nabla f_i(\mathbf{x}_{j-1})$.

The protocol is terminated upon reaching a predetermined number of epochs.

SGD guarantees fast convergence under different assumptions (e.g., when the $f_i$-s are convex functions), but requires random access to individual examples, which is inefficient when training on large datasets stored in the cloud. The CorgiPile algorithm (see Algorithm 1) was proposed in (Xu et al., 2022a) as an alternative to SGD with random access, improving efficiency by accessing blocks of examples together. This algorithm assumes that the data is horizontally (i.e., row-wise) sharded across $N$ blocks of size $b$, resulting in a dataset size of $m = Nb$. CorgiPile operates by iteratively picking $n$ blocks randomly from the dataset to fill a buffer $S$, shuffling the buffer, and running SGD on the examples in the buffer.

In alignment with Xu et al. (2022a), we presume a read-write buffer $S$ with random access, capable of containing up to $nb$ examples simultaneously, namely $|S| = nb$. We introduce a new algorithm, Corgi[2] (see Algorithm 3), which improves the convergence guarantees of CorgiPile while maintaining efficient data access, at the cost of adding an efficient offline step that reorganizes the data before the training starts. More specifically, the Corgi[2] algorithm initially executes an offline, slightly modified version of CorgiPile, repurposed to redistribute examples among blocks in a manner that minimizes

---

**Algorithm 1** CorgiPile

---

1: **input:** Blocks $\{B_i\}_{i=1}^N$, each of size $b$, with a total of $Nb$ examples;
   Model parameterized by $\mathbf{x}$; number of epochs $\mathcal{T} \geq 1$; buffer size $n \geq 1$.
2: **for** $t = 1, \ldots, \mathcal{T}$ **do**:
3:     Randomly pick $n$ blocks (i.i.d without replacement).
4:     Shuffle the indices of the resulting buffer, obtaining permutation $\Psi_t$ over $\{1, \ldots, nb\}$.
5:     $\mathbf{x}_0^{(t)} \leftarrow \mathbf{x}_{nb}^{(t-1)}$
6:     **for** $j = 1, \ldots, nb$ **do**:
7:        $\mathbf{x}_j^{(t)} = \mathbf{x}_{j-1}^{(t)} - \eta_t \nabla f_{\Psi_t(j)}\left(\mathbf{x}_{j-1}^{(t)}\right).$
8:     **end for**
9: **end for**
10: **return** $\mathbf{x}_{nb}^{(\mathcal{T})}$

---

block variance (see Algorithm 2). Subsequently, it runs the original online CorgiPile algorithm (Algorithm 1) on the preprocessed dataset.

---

**Algorithm 2** OfflineCorgiShuffle

---

1: **input:** Blocks $\{B_i\}_{i=1}^N$, each of size $b$, with a total of $Nb$ examples; buffer size $n \geq 1$.
2: **for** $l = 1, \ldots, N/n$ **do**:
3:     Randomly pick $n$ blocks (i.i.d with replacement), and fill the buffer $S$.
4:     **for** $j = 1, \ldots, n$ **do**:
5:        Randomly pick $b$ examples from the $nb$ examples in $S$ (i.i.d with replacement).
6:        Create a new block $\widetilde{B}_{l,j}$ with the chosen tuples.
7:     **end for**
8: **end for**
9: **return** $\{\widetilde{B}_{l,j}\}$ for $j \in \{1, \ldots, n\}$ and $l \in \{1, \ldots, \frac{N}{n}\}$.

---

**Algorithm 3** The Corgi$^2$ Algorithm

---

1: **Input:** blocks $\{B_i\}_{i=1}^N$, each of size $b$, with a total of $Nb$ examples; a model parameterized by $\mathbf{x}$; the number of epochs $\mathcal{T} \geq 1$; and a buffer size $n \geq 1$.
2:     Execute the OfflineCorgiShuffle procedure to obtain the shuffled blocks $\{\widetilde{B}_{l,j}\}$.
3:     Apply the CorgiPile method to the shuffled data $\{\widetilde{B}_{l,j}\}$.
4: **Output:** The updated model parameters $\mathbf{x}_{nm}^{(\mathcal{T})}$ after $\mathcal{T}$ epochs.

---

We note that the additional cost in terms of time and number of data access queries due to using Corgi$^2$ is minimal, compared to the original CorgiPile algorithm (see Section 3.3). However, a naive implementation of Corgi$^2$ doubles the cost of storage, which may be significant for large datasets. That said, we observe that with a small modification to the offline step of Corgi$^2$, namely by selecting blocks i.i.d. *without* replacement, it is possible derive a variant of Corgi$^2$ that reorganizes the data in-place, and thus consumes no extra storage. While this variant is harder to analyze theoretically, it obtains similar (and often better) performance in practice.

## 3 THEORY

In this section, we analyze the convergence time of Corgi$^2$ under some assumptions, and show that the cost of the additional offline step (in terms of data access) is relatively small.

### 3.1 NOTATIONS AND ASSUMPTIONS

We make the following assumptions about the loss functions and the examples in the dataset. These assumptions are taken from the analysis in (Xu et al., 2022a).

1. $f_i(\cdot)$ are twice continuously differentiable.
2. $L$-Lipschitz gradient: $\exists L \in \mathbb{R}_+, \|\nabla f_i(\mathbf{x}) - \nabla f_i(\mathbf{y})\| \leq L\|\mathbf{x} - \mathbf{y}\|$ for all $i \in [m], \forall \mathbf{x}, \mathbf{y}$.
3. $L_H$-Lipschitz Hessian matrix: $\|H_i(\mathbf{x}) - H_i(\mathbf{y})\| \leq L_H\|\mathbf{x} - \mathbf{y}\|$ for all $i \in [m], \forall \mathbf{x}, \mathbf{y}$.
4. Bounded gradient: $\exists G \in \mathbb{R}_+, \|\nabla f_i(\mathbf{x})\| \leq G$ for all $i \in [m], \forall \mathbf{x}$.
5. Bounded Variance: $\frac{1}{m}\sum_{i=1}^{m}\|\nabla f_i(\mathbf{x}) - \nabla F(\mathbf{x})\|^2 \leq \sigma^2, \forall \mathbf{x}$.

A crucial aspect in examining the original CorgiPile algorithm (Xu et al., 2022a) is the constraint on the block-wise gradient variance. Intuitively, this variance is minimized when the gradient of each block $\nabla f_{B_l}$ closely approximates the gradient over the entire data $\nabla F$. More formally,

$$\frac{1}{N}\sum_{l=1}^{N}\left\|\nabla f_{B_l}(\mathbf{x}) - \nabla F(\mathbf{x})\right\|^2 \leq h_D \frac{\sigma^2}{b}, \tag{2}$$

where $\nabla f_{B_l} := \frac{1}{b}\sum_{i \in B_l}\nabla f_i$ is the mean gradient in the block, and $h_D$ represents a constant that characterizes the variability of this block-wise gradient.

It is important to note that the inequality in Equation 2 always holds for $h_D = b$, and the value of $h_D$ will be larger when each block exhibits greater homogeneity. For instance, consider an image dataset in which each block comprises sequential frames from a single video. In this case, the dataset would exhibit a high degree of homogeneity. Conversely, a well-shuffled dataset, wherein blocks possess highly similar distributions, could yield $h_D$ values approaching 1. There are even edge cases, such as a perfectly balanced dataset with identical gradients across all blocks, that would result in $h_D = 0$.

## 3.2 MAIN RESULT

We start by showing that after the offline step of Corgi$^2$ (i.e., after running OfflineCorgiShuffle), the block-wise variance decreases compared to the variance of the original blocks.

**Theorem 1.** *Consider the execution of* OfflineCorgiShuffle *(refer to Algorithm 2) on a dataset characterized by a variance bound $\sigma^2$, a block-wise gradient variance parameter $h_D$, $N$ blocks each containing $b$ examples, and a buffer size $nb$. For all $\mathbf{x}$, the following inequality holds:*

$$\mathbb{E}\left[\frac{1}{N}\sum_{l=1}^{N}\left\|\nabla f_{\widetilde{B}_l}(\mathbf{x}) - \nabla F(\mathbf{x})\right\|^2\right] \leq h_D' \frac{\sigma^2}{b}, \tag{3}$$

*where $h_D' = 1 + \left(\frac{1}{n} - \frac{1}{nb}\right) h_D$, and $f_{\widetilde{B}_l}(\mathbf{x})$ denotes the average of examples in $\widetilde{B}_l$, the l-th block generated by the* OfflineCorgiShuffle *algorithm.*

The ratio between $h_D'$ and $h_D$ can thus be expressed as,

$$\frac{h_D'}{h_D} = \frac{1}{h_D} + \frac{b-1}{nb}. \tag{4}$$

This ratio is less than 1 whenever the following condition is satisfied:

$$h_D > \frac{nb}{(n-1)b - 1}. \tag{5}$$

This condition is trivially met when $n$ and $b$ assume realistic values (e.g., it holds if $n > 2$ and $b > 2$). It is worth noting that larger values of $h_D$ amplify the ratio, aligning with the intuitive understanding that datasets initially divided into more homogeneous blocks would undergo a more substantial reduction in variance during the offline shuffle phase.

Blockwise variance significantly impacts the disparity between the anticipated distribution of a buffer comprising $n$ blocks and the overall dataset distribution. In turn, this lowers the convergence rate, bringing it closer to that of random access SGD during training. Further elaboration on this relationship is provided in Theorem 2.

**Theorem 1 Proof Sketch:** Since Offline Corgi works on each generated block $\tilde{B}_l$ independently, we analyze a single iteration of the algorithm. We focus on the expression,

$$\mathrm{V}_{S,\widetilde{B}_l}\left(\nabla f_{\widetilde{B}_l}\right) := \frac{1}{N}\sum_{l=1}^{N}\left\|\nabla f_{\widetilde{B}_l}(\mathbf{x}) - \nabla F(\mathbf{x})\right\|^2.$$

where $S$ is a vector representation of the buffer, $\widetilde{B}_l$ represents the block created from it by uniformly sampling from the buffer and $l$ is a uniformly sampled index. This is a measure of variance that generalizes scalar variance, expressed as a scalar rather than a matrix. This measure, has similar properties to standard variance such as $\mathrm{V}\left(\alpha X\right) = \alpha^2\mathrm{V}\left(X\right)$ and law of total variance (see Appendix A for a full discussion). Thus we can decompose the left hand side of the theorem equation using the law of total variance:

$$\frac{1}{N}\sum_{l=1}^{N}\left\|\nabla f_{\widetilde{B}_l}(\mathbf{x}) - \nabla F(\mathbf{x})\right\|^2 = \mathrm{V}_{S,\widetilde{B}_l}\left(\nabla f_{\widetilde{B}_l}(\mathbf{x})\right) = \underbrace{\mathrm{V}_S\left(\mathbb{E}_{\widetilde{B}_l}\left[\nabla f_{\widetilde{B}_l}|S\right]\right)}_{(i)} + \underbrace{\mathbb{E}_S\left[\mathrm{V}_{\widetilde{B}_l}\left(\nabla f_{\widetilde{B}_l}|S\right)\right]}_{(ii)}$$

When $S$ is fixed, each $\widetilde{B}_l$ is an unbiased i.i.d selection of $b$ examples from it.

Thus, in $(i)$, given fixed $S$ we have $\mathbb{E}_{\widetilde{B}_l}\left[\nabla f_{\widetilde{B}_l}|S\right] = \nabla f_S := \frac{1}{nb}\sum_{i\in S}\nabla f_i$, i.e., the average gradient in the buffer. In turn, since $S$ is an i.i.d sampling of $n$ blocks, the variance of its average is equal to $\frac{1}{n}$ of the variance of sampling the average of a single block, which gives us

$$(i): \mathrm{V}_S\left(\mathbb{E}_{\widetilde{B}_l}\left[\nabla f_{\widetilde{B}_l}|S\right]\right) = \mathrm{V}_S\left(\nabla f_S|S\right) = \frac{1}{n}\mathrm{V}_i\left(\nabla f_{B_i}\right) \leq \frac{1}{n}h_D\frac{\sigma^2}{b}.$$

Where $B_i$ is the $i^{\text{th}}$ block, before applying the Offline Corgi.

For term $(ii)$, we apply Bienaymé's identity and use the fact that averaging $b$ i.i.d. elements decreases the variance by a factor of $\frac{1}{b}$ compared to the variance of sampling a single element. Given that, and letting $i$ be an index selected uniformly from $1, ..., bn$, we observe that $\mathbb{E}_S\left[\mathrm{V}_{\widetilde{B}_l}\left(\nabla f_{\widetilde{B}_l}|S\right)\right] = \frac{1}{b}\mathbb{E}_S[\mathrm{V}_i\left(S_i|S\right)]$. This expression can be decomposed to,

$$\mathbb{E}_S\left[\mathrm{V}_{\widetilde{B}_l}\left(\nabla f_{\widetilde{B}_l}|S\right)\right] = \frac{1}{b}\mathbb{E}_S[\mathrm{V}_i\left(S_i|S\right)] = \frac{1}{b}\left(\underbrace{\mathrm{V}_i\left(S_i\right)}_{I} - \underbrace{\mathrm{V}_S\left(\mathbb{E}_i[S_i|S]\right)}_{II}\right).$$

Here $(I)$ is the variance of sampling one element from the buffer, before the buffer itself is known. Since every example from the dataset has the same probability of being the $i^{\text{th}}$ example in $S$, this variance is equal to the variance of the dataset itself, which is bounded by $\sigma^2$.

Moreover, $(II)$ is the variance of the average of $S$, and exactly like in $(i)$, it equals the pre-shuffle blockwise variance. Put together,

$$(ii): \mathbb{E}_S\left[\mathrm{V}_{\widetilde{B}_l}\left(\nabla f_{\widetilde{B}_l}|S\right)\right] = \frac{1}{b}\mathbb{E}_S[\mathrm{V}_i\left(S_i|S\right)] \leq \left(1 - \frac{1}{nb}\right)\frac{\sigma^2}{b}$$

Combining the bounds for $(i)$ and $(ii)$ yield the result. For a full proof see Appendix B.

### 3.2.1 CONVERGENCE RATE ANALYSIS

The convergence rate of CorgiPile and $\text{Corgi}^2$ is expected to be slower (in terms of epochs) than that of random access SGD, especially when the individual buffers significantly differ from the distribution of the dataset as a whole. Specifically, larger values of $\frac{n}{N}$ would guarantee faster convergence time as more of the dataset is shuffled together in each iteration; and higher values of $h_D$ would hurt convergence time as the variance in each iteration is increased.

In the following theorem we revisit the convergence rate upper bound associated with CorgiPile and establish the extent to which our approach contributes to its reduction.

**Theorem 2.** *Suppose that $F(\mathbf{x})$ are smooth and $\mu$-strongly convex function. Let $T = \mathcal{T}nb$ be the total number of examples seen during training, where $\mathcal{T} \geq 1$ is the number of buffers iterated. Choose the learning rate to be $\eta_t = \frac{6}{bn\mu(t+a)}$ where $a \geq \max\left\{\frac{8LG+24L^2+28L_HG}{\mu^2}, \frac{24L}{\mu}\right\}$. Then, Corgi$^2$, has the following convergence rate in the online stage for any choice of $\mathbf{x}_0$,*

$$\mathbb{E}[F\left(\bar{\mathbf{x}}_{\mathcal{T}}\right) - F(\mathbf{x}^*)] \leq (1-\alpha)h'_D\sigma^2\frac{1}{T} + \beta\frac{1}{T^2} + \gamma\frac{(Nb)^3}{T^3}, \tag{6}$$

*where $\bar{\mathbf{x}}_{\mathcal{T}} = \frac{\sum_t (t+a)^3\mathbf{x}_t}{\sum_t (t+a)^3}$, and*

$$\alpha := \frac{n-1}{N-1}, \beta := \alpha^2 + (1-\alpha)^2(b-1)^2, \gamma := \frac{n^3}{N^3}.$$

A full proof is provided in Appendix B, but boils down to wrapping the convergence rate proved for CorgiPile in an expectation over the randomness of Offline Corgi and updating the expression accordingly. The convergence rate for CorgiPile in the same setting is,

$$\mathbb{E}[F\left(\bar{\mathbf{x}}_{\mathcal{T}}\right) - F(\mathbf{x}^*)] \leq (1-\alpha)h_D\sigma^2\frac{1}{T} + \beta\frac{1}{T^2} + \gamma\frac{N^3b^3}{T^3}, \tag{7}$$

Observe that the difference between the methods is expressed in the replacement of the blockwise variance parameter $h_D$ with $h'_D$. As is shown in Theorem 1, $h'_D$ will be lower in practically all cases. Here we see that $h'_D$ controls the convergence rate, as it linearly impacts the leading term $\frac{1}{T}$.

We now have a good grasp on the guiding principles of when Corgi$^2$ can be expected to converge significantly faster than CorgiPile. Specifically, when the original blocks are homogeneous, we expect that $h_D = \Theta(b)$, in which case Corgi$^2$ will improve the convergence rate by a factor of $1/n$ (where $n$ is the number of blocks in the buffer). On the other hand, when data is already shuffled, we expect that $h_D = \Theta(1)$, in which case Corgi$^2$ will not give a significant improvement, and can even hurt convergence. In the following subsection we show that, Corgi$^2$ improves data efficiency by a factor of $1/b$ over full shuffle.

## 3.3 COMPLEXITY ANALYSIS

Table 1: Number of data access queries for the different shuffling methods.

| Algorithm | # Offline queries | # Online queries | Total |
|---|---|---|---|
| Random Access | – | $\mathcal{T}m$ | $\mathcal{T}m$ |
| Shuffle-Once | $m + m/b$ | $\mathcal{T}m/b$ | $m + (\mathcal{T}+1)m/b$ |
| CorgiPile | – | $\mathcal{T}m/b$ | $\mathcal{T}m/b$ |
| Corgi$^2$ | $2m/b$ | $\mathcal{T}m/b$ | $(\mathcal{T}+2)m/b$ |

Previous sections demonstrated that Corgi$^2$ improves the convergence rate of CorgiPile by a significant factor. We now turn our attention to quantifying the increase in query complexity that Corgi$^2$ induces.

In our study, we conceptualize the storage system as managing *chunks* comprising $b$ examples, wherein each input/output (IO) operation pertains to an entire chunk. Consequently, the cost incurred for accessing a single example or all $b$ examples within the same chunk remains identical. This unpretentious modeling aptly delineates the cost structure associated with cloud-based data storage, given that providers such as Amazon Web Services (AWS) impose a fixed fee for each object access, irrespective of the object's size (Amazon, 2023; Microsoft, 2023). Bearing this model in mind, we can evaluate various shuffling algorithms by employing the elementary metric of *number of data access operations*.

In Table 1, we compare CorgiPile and Corgi$^2$ to more traditional approaches such as random access SGD and shuffling the data one time. Standard SGD simply requires $\mathcal{T}m$ queries, where $\mathcal{T}$ is the

number of training epochs. On the other hand, the Shuffle-Once approach (see (Safran & Shamir, 2020)) necessitates $m + (\mathcal{T} + 1)m/b$ queries: $m$ read operations for one example each, accompanied by $m/b$ write operations to store the data in shuffled chunks. Then, $\mathcal{T}m/b$ read operations to fetch full chunks during training.

Meanwhile, CorgiPile costs only $\mathcal{T}m/b$ queries in total, since each chunk is read exactly once in each epoch. Corgi$^2$ incurs an additional cost of $2m/b$ queries (read + write) in the preceding offline phase. Thus, up to a small constant factor, Corgi$^2$ demands the same number of queries as CorgiPile.

Lastly, we note that our metric expresses query complexity and not time complexity, since realistic executions of shuffle methods rely heavily on parallelization techniques, which might be limited by factors such as software implementation and the throughput limits of the storage system. The Corgi$^2$ algorithm itself imposes no bottlenecks on parallelization, meaning that it should enjoy similar benefits to run time complexity as those of the other methods analyzed here.

## 4  EXPERIMENTS

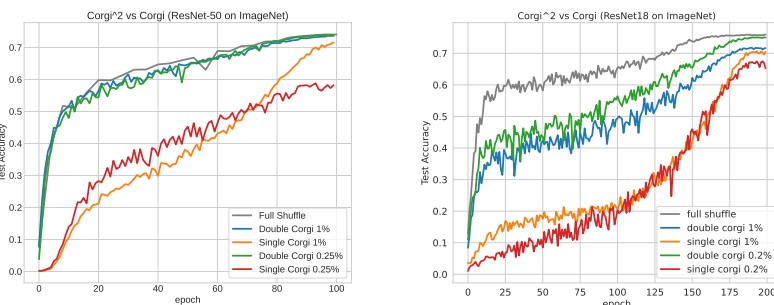

Figure 3: Training performance comparison between full shuffle, CoriPile and Corgi$^2$. Left: Test accuracy for ResNet-50 on ImageNet, Right: Test accuracy for CIFAR-100 on ResNet-18.

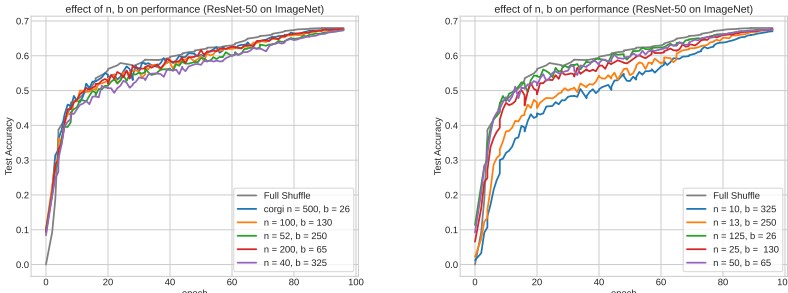

Figure 4: Training performance of ResNet50 on ImageNet as in Figure 5 when varying block sizes and number of blocks in each buffer. (Left) Buffer size set to 0.25%. Right: Buffer size set to 1%.

Thus far, we have observed that Corgi$^2$ offers substantial improvements in terms of theoretical guarantees over the original CorgiPile method, particularly in the context of homogeneously sharded data. In this section, we aim to empirically investigate the performance of Corgi$^2$ for such datasets and compare it to CorgiPile and full data shuffle, where the data is randomly reshuffled every epoch. CorgiPile has been shown to achieve parity with a full shuffle for buffer sizes of about 2% or more, but noticeably dip in performance below that threshold. Our tests explore buffer sizes as low as 0.25%, which are viable with common hardware for datasets of even hundreds of terabytes.

Figure 4 displays an experiment on two open source image classification tasks, specifically ResNet-18 (He et al., 2016) on CIFAR-100, and ResNet-50 on ImageNet (Deng et al., 2009). It uses a Corgi$^2$ implementation written in the pytorch framework to perform the shuffle. We observe that while the ImageNet experiment shows *corgii* achieving performance parity with full shuffle, this isn't

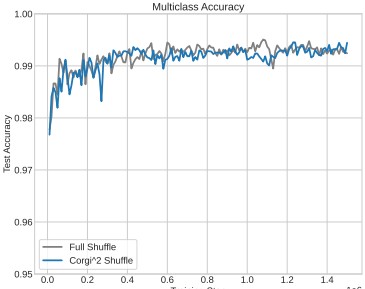 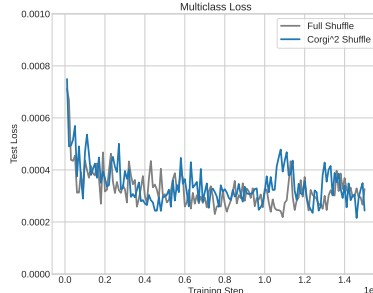

Figure 5: Training performance comparison (accuracy and loss) between the shuffle methods on a proprietary image classification model.

replicated in CIFAR100, ostensibly a less complex dataset. We believe this to be artifact of the buffer size being small compared to *the number of classes* in CIFAR100 - see appendix C for further details.

Figure 4 demonstrates the tradeoff between block size and buffer size - the same block sizes result in considerably better performance in the right plot (with a buffer size of 1%) than the left one (with a buffer size of 0.25%). While smaller blocks will boost the training performance for a given buffer size, a careful balance must be struck in order to maintain good data access efficiency.

Figure 5 shows an experiment for multiclass classification over a proprietary computer vision dataset of 27TB (obscured for the duration of the peer review process). The model is tasked with classifying crops of objects taken out of real-world images, which in this experiment has been trained on 27TB of such data. The comparison is between a version of the dataset which was fully shuffled, and a version shuffled by a distributed Corgi$^2$ implementation where each node had enough memory for about 0.25% of the data.

## 5 DISCUSSION

Corgi$^2$ is a straightforward algorithm that can be easily modified for various purposes. Following are brief discussions of some modifications of interest:

1.  **Repeated offline shuffles:** One intuitive way to modify Corgi$^2$ is to repeat the offline phase (2) multiple times, to further reduce block variance before the online phase. This would incur a cost in query complexity, as outlined in table 1, but each such repetition would lower the parameter $h_D$ according to equation 4 (and consequently improve the convergence rate as described in theorem 2). Thus, the magnitude of the reduction diminishes exponentially with each further repetition, but query complexity increases linearly. While scenarios where this modification is useful may very well exists, it will usually be more cost effective to boost performance by increasing the number of blocks in the buffer.

2.  **Sampling without replacement:** As mentioned in section 2, the motivation for sampling with replacement in Corgi$^2$ was to streamline the theoretical analysis, despite understanding that sampling without replacement is preferred in real world applications. Empirically, most experiments in section 4 were repeated in both ways, with no discernible differences.

3.  **Overwriting blocks to conserve storage:** Section 2 mentions a modification that avoids doubling the storage requirements during execution of Corgi$^2$. This modification will simply have Algorithm 2 delete each block it finishes reading, thus maintaining the number of blocks. We note that this modification will result in permanent data loss, unless combined with sampling without-replacement.

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

# A  VARIANCE

In Theorem 1 we employ a generalization (Kocherlakota & Kocherlakota, 2004) of scalar variance that can apply to vectors of arbitrary dimensions. Let $X \in \mathbb{R}^d$ some random variable, and let $\mu = \mathbb{E}[X]$. then:

$$V(X) = \mathbb{E}\left[\|X - \mu\|_2\right] = \mathbb{E}\left[(X - \mu)^T(X - \mu)\right] \tag{8}$$

This diverges from the more common definition of variance:

$$V(X) = V\left(\begin{pmatrix} x_1 \\ \vdots \\ x_n \end{pmatrix}\right) = \mathbb{E}\left[(X - \mu)(X - \mu)^T\right]$$

Equation 8 is a generalization of variance in the sense that when $d = 1$ we get the standard variance definition for scalar random variables.

This appendix specifies and proves all properties of this measure which are used in Section B.1.

**1:** $V(X) = \mathbb{E}\left[X^T X\right] - \mu^T \mu$

$$V(X) = \mathbb{E}\left[(X - \mu)^T(X - \mu)\right] = \mathbb{E}[X^T X] - \mathbb{E}[X^T \mu] - \mathbb{E}[\mu^T X] + \mu^T \mu$$
$$= \mathbb{E}[X^T X] - \mu^T \mu - \mu^T \mu + \mu^T \mu = \mathbb{E}[X^T X] - \mu^T \mu$$

**2:** $V(aX) = a^2 V(X)$

$$V(aX) = \mathbb{E}\left[a(X - \mu)^T a(X - \mu)\right] = a^2 \mathbb{E}\left[(X - \mu)^T(X - \mu)\right] = a^2 V(X)$$

**3:** $V(X + Y) = V(X) + V(Y) + \text{COV}(X, Y) + \text{COV}(Y, X)$

Where $\text{COV}(X, Y)$ is the cross covariance between $X$ and $Y$, defined as $\mathbb{E}\left[[(X - \mu_X)^T(Y - \mu_Y)]\right]$

$$
\begin{aligned}
V(X + Y) &= \mathbb{E}\left[[(X + Y - \mu_X - \mu_Y)^T(X + Y - \mu_X - \mu_Y)]\right] \\
&= \mathbb{E}\left[X^T X - X^T \mu_X - \mu_X^T X + \mu_X^T \mu_X\right] + \mathbb{E}\left[Y^T Y - Y^T \mu_Y - \mu_Y^T Y + \mu_Y^T \mu_Y\right] \\
&\quad + \mathbb{E}\left[X^T Y - X^T \mu_Y - \mu_X^T Y + \mu_X^T \mu_Y\right] + \mathbb{E}\left[Y^T X - Y^T \mu_X - \mu_Y^T X + \mu_Y^T \mu_X\right] \\
&= \mathbb{E}\left[X^T X\right] - \mu_X^T \mu_X - \mu_X^T \mu_X + \mu_X^T \mu_X + \mathbb{E}\left[Y^T Y\right] - \mu_Y^T \mu_Y - \mu_Y^T \mu_Y + \mu_Y^T \mu_Y \\
&\quad + \mathbb{E}\left[(X - \mu_X)^T(Y - \mu_Y)\right] + \mathbb{E}\left[(Y - \mu_Y)^T(X - \mu_X)\right] \\
&= \mathbb{E}\left[X^T X\right] - \mu_X^T \mu_X + \mathbb{E}\left[Y^T Y\right] - \mu_Y^T \mu_Y + \text{COV}(X, Y) + \text{COV}(Y, X) \\
&= V(X) + V(Y) + \text{COV}(X, Y) + \text{COV}(Y, X)
\end{aligned}
$$

-

**4:** $V(X) = V(\mathbb{E}[X|Y]) + \mathbb{E}[V(X|Y)]$ (Law Of Total Variance)

First,

$$
\begin{aligned}
\mathbb{E}[X^T X] &= \mathbb{E}\left[\mathbb{E}[X^T X|Y]\right] = \mathbb{E}[X^T X] = \mathbb{E}\left[\mathbb{E}[X^T X|Y] - \mathbb{E}^2[X|Y] + \mathbb{E}^2[X|Y]\right] \\
&= \mathbb{E}[V(X|Y) + \mathbb{E}^2[X|Y]]
\end{aligned}
$$

Then,

$$
\begin{aligned}
V(X) &= -\mathbb{E}^2[X] + \mathbb{E}[X^T X] = -\mathbb{E}^2[\mathbb{E}[X|Y]] + \mathbb{E}[\mathbb{E}^2[X|Y] + V(X|Y)] \\
&= -\mathbb{E}^2[\mathbb{E}[X|Y]] + \mathbb{E}[\mathbb{E}^2[X|Y]] + \mathbb{E}[V(X|Y)] \\
&= V(\mathbb{E}[X|Y]) + \mathbb{E}[V(X|Y)]
\end{aligned}
$$

## B  PROOFS

### B.1  PROOF OF THEOREM 1

**Theorem 1.** *Consider the execution of* OfflineCorgiShuffle *(refer to Algorithm 2) on a dataset characterized by a variance bound $\sigma^2$, a block-wise gradient variance parameter $h_D$, $N$ blocks each containing $b$ examples, and a buffer size $nb$. For all* **x**, *the following inequality holds:*

$$\mathbb{E}\left[\frac{1}{N}\sum_{l=1}^{N}\left\|\nabla f_{\widetilde{B}_l}(\mathbf{x}) - \nabla F(\mathbf{x})\right\|^2\right] \le h_D' \frac{\sigma^2}{b}, \tag{9}$$

*where $h_D' = 1 + \left(\frac{1}{n} - \frac{1}{nb}\right)h_D$, and $f_{\widetilde{B}_l}(\mathbf{x})$ denotes the average of examples in $\widetilde{B}_l$, the $l$-th block generated by the* OfflineCorgiShuffle *algorithm.*

#### B.1.1  PROOF

First we establish notation that will be used throughout the proof:

- $f_i$, the loss of $x$ over the $i$-th element of the dataset, or simply "the $i$-th function". $\nabla f_i$ is $\in [1, ..., Nb]$

- $B_i = \begin{pmatrix} f_{ib} \\ \vdots \\ f_{(i+1)b} \end{pmatrix}$ is the $i$-th block

- $f_{B_l} = \sum_{i=lb}^{b}(l+1)b$ is the average of functions in the $l$-th block.

- For each of the $\frac{N}{n}$ iteration of OfflineCorgi (Algorithm 2):

  - $S = \begin{pmatrix} B_{i_1} \\ \vdots \\ B_{i_n} \end{pmatrix}$ is a random vector composed of $n$ uniform i.i.d selections of blocks. $S_i$ is the $i$-th row of $S$, corresponding to a single function.

  - $\widetilde{B}_l = \begin{pmatrix} S_{i_1} \\ \vdots \\ S_{i_b} \end{pmatrix}$ if the $l$-th of $n$ new blocks created this round, composed of $b$ uniform i.i.d selections of rows from $S$.

  - $f_{\widetilde{B}_l}$ is the block average for new blocks, and $f_S$ is the buffer average, defined similarly to $f_{B_l}$ above.

**Lemma 2.** *Let $\{\widetilde{B}_l\}_{l=1}^{nj}$ be the set of blocks created by $j$ iterations of the main loop in OfflineCorgi (Algorithm 2). Let $i$ be an index uniformly selected from $[1, .., jn]$. Then the **r.v** $\widetilde{B}_i$ has the same distribution, for all $j$.*

*Proof.* Denote $\mathcal{B}^j = (B_{j'n}, \ldots B_{(j'+1)n-1})$ the blocks generated by the $j'$ iteration. Notice that $\mathcal{B}^1, \ldots, \mathcal{B}^j$ are i.i.d. random variables. Define the random variable $x \sim [1, ..., j]$ and $y \sim [1, ..., n]$ such that $i = xn + y$. Then, $\widetilde{B}_i = \mathcal{B}_y^x$, and the required follows.  □

Assume that there is exactly one iteration of OfflineCorgi. Using the above notation, we can rewrite the theorem as:

$$\mathrm{V}_{S,\widetilde{B},j}\left(\nabla f_{\widetilde{B}_j}\right) \le h_D' \frac{\sigma^2}{b}$$

where $j$ is an index sampled uniformly from $[1, .., n]$, and $V$ is the generalized scalar variance discussed in Appendix A. According to Lemma 2, proving this is sufficient to prove the general case of Theorem 1. Using the law of total variance:

$$\frac{1}{N}\sum_{l=1}^{N}\left\|\nabla f_{\widetilde{B}_l}(\mathbf{x}) - \nabla F(\mathbf{x})\right\|^2 = \mathrm{V}_{S,\widetilde{B}_l}\left(\nabla f_{\widetilde{B}_l}(\mathbf{x})\right) = \underbrace{\mathrm{V}_S\left(\mathbb{E}_{\widetilde{B}_l}\left[\nabla f_{\widetilde{B}_l}|S\right]\right)}_{(i)} + \underbrace{\mathbb{E}_S\left[\mathrm{V}_{\widetilde{B}_l}\left(\nabla f_{\widetilde{B}_l}|S\right)\right]}_{(ii)}$$

**(i):**

When $S$ is fixed, each $\widetilde{B}$ is a uniform i.i.d selection of $b$ functions from $S$. Let $\begin{pmatrix} z_1 \\ \vdots \\ z_{nb} \end{pmatrix}$ be a random vector s.t $z_i$ is a random variable for the number of times $S_i$ has been selected in this process to a given $\widetilde{B}$. Then $\widetilde{B_l}$ can be written as $ZS$, where $Z$ is a diagonal matrix with $Z_{i,i} = z_i$. The resulting r.v is a multinomial distribution with $b$ experiments and $nb$ possible results per experiment, each with an equal probability $\frac{1}{nb}$. Thus:

$$\mathbb{E}_{\widetilde{B_l}}\left[\nabla f_{\widetilde{B_l}}|S\right] = \nabla\left(\frac{1}{b}\sum_{i=1}^{nb}(\mathbb{E}_Z[ZS])_i\right)$$

$$= \nabla\left(\frac{1}{b}\sum_{i=1}^{nb}\mathbb{E}[z_i]S_i\right) = \nabla\left(\sum_{i=1}^{nb}\frac{1}{n}S_i\right)$$

$$= \nabla\left(\frac{1}{nb}\sum_{i=1}^{nb}s_i\right) = \nabla f_S$$

We now have:

$$\mathrm{V}_S\left(\mathbb{E}_{\widetilde{B_l}}\left[\nabla f_{\widetilde{B_l}}|S\right]\right) = \mathrm{V}_S\left(\nabla f_S\right)$$

For a given $\begin{pmatrix} B_{i_1} \\ \vdots \\ B_{i_n} \end{pmatrix}$, we have $\nabla f_S = \frac{1}{n}\sum_{m=1}^{n}\nabla f_{B_{i_m}}$ where $i_1, ..., i_n$ are the $n$ blocks selected for $S$. Thus:

$$\mathrm{V}_S\left(\nabla f_S\right) = \mathrm{V}_S\left(\frac{1}{n}\sum_m \nabla f_{B_{i_m}}\right) = \frac{1}{n^2}\mathrm{V}\left(\sum_m \nabla f_{B_{i_m}}\right) \le \frac{1}{n}h_D\frac{\sigma^2}{b}$$

Where the inequality is due to the $n$ block selections being i.i.d and the upper bound on block variance is described in assumption 5 in Section 3.1.

**(ii):**

Let i $i$ be a uniformly sampled index in the range $[1, ..., nb]$. For a fixed $S$, define the *sampling variance* to be

$$\mathrm{V}_i\left(S_i|S\right) = \mathbb{E}_i\left[S_i^T S_i\right] - \nabla f_S^T \nabla f_S = \frac{1}{nb}\sum_i\langle s_i, s_i\rangle - \frac{1}{(nb)^2}\left(\sum_i\langle s_i, s_i\rangle + \sum_{i\ne j}\langle s_i, s_j\rangle\right)$$

This, in other words, is the variance of uniformly sampling a function from $S$.

We wish to find $\mathrm{V}\left(\widetilde{B}|S\right)$. We define random variables as we did in $(i)$ and apply Bienaymé's identity:

$$\mathrm{V}_i\left(\nabla f_{\widetilde{B_j}}|S\right) = \frac{1}{b^2}V\left(\sum_{i=1}^{nb} z_i s_i\right)$$

$$= \frac{1}{b^2}\left(\sum_i V(z_i)\langle s_i, s_i\rangle + \sum_{i\neq j}\mathrm{COV}(z_i, z_j)\langle s_i, s_j\rangle\right)$$

$$= \frac{1}{b^2}\left(\sum_i b\frac{1}{nb}(1 - \frac{1}{nb})\langle s_i, s_i\rangle + \sum_{i\neq j}\left(-\frac{b}{n^2 b^2}\right)\langle s_i, s_j\rangle\right)$$

$$= \frac{1}{b^2}\left(\sum_i \frac{1}{n}\langle s_i, s_i\rangle - \sum_i \frac{1}{n^2 b}\langle s_i, s_j\rangle - \sum_{i\neq j}\frac{1}{n^2 b}\langle s_i, s_j\rangle\right)$$

$$= \frac{1}{b}\left(\frac{1}{nb}\sum_i \langle s_i, s_i\rangle - \frac{1}{n^2 b^2}\left(\sum_i \langle s_i, s_i\rangle + \sum_{i\neq j}\langle s_i, s_j\rangle\right)\right)$$

$$= \frac{1}{b}\mathrm{V}_i\left(S_i|S\right)$$

We now have:

$$\mathbb{E}_S\left[\mathrm{V}_{\widetilde{B_l}}\left(\nabla f_{\widetilde{B_l}}|S\right)\right] = \frac{1}{b}\mathbb{E}_S[\mathrm{V}_i\left(S_i|s\right)]$$

We further decompose this expression by a second application of the law of total variance:

$$\mathbb{E}_S[\mathrm{V}_i\left(S_i\right)|S] = \underbrace{\mathrm{V}_i\left(S_i\right)}_{I} - \underbrace{\mathrm{V}_S\left(\mathbb{E}_i[S_i|S]\right)}_{II}$$

**II**: $\mathbb{E}_i[S_i|S]$ is the expected value of sampling a function from a fixed $S$, which is simply $\nabla f_S$. Duplicating the calculation done for $(i)$, $\mathrm{V}_S\left(\nabla f_S\right) \leq h_D\frac{\sigma^2}{b}$.

**I**: When $S$ is not fixed, it is a uniform i.i.d selection of blocks for $[B_1, ..., B_N]$. Let $\begin{pmatrix} z_1 \\ \vdots \\ z_N \end{pmatrix}$ be a random vector s.t $z_i$ is a random variable for the number of times $B_i$ has been selected by this process for a given $S$. Then $S$ can be written as $ZB$ where $Z$ is a diagonal matrix with $Z_{i,i} = z_i$, and $B = \begin{pmatrix} B_1 \\ \vdots \\ B_N \end{pmatrix}$.

Since $S$ is a multinomial with $n$ experiments and $N$ possible results with probability $\frac{1}{N}$,

$$\forall i, k \geq 0 \quad \Pr(z_i = k) = \binom{n}{k}\left(\frac{1}{N}\right)^k\left(\frac{N-1}{N}\right)^{n-k}$$

Let $f$ be any function in some block $B_j$. Then:

$$\Pr\left(S_i = f\right) = \sum_{k=1}^{n} \Pr\left(S_i = f | z_j = k\right) \Pr\left(z_j = k\right)$$

$$= \sum_{k=1}^{n} \binom{n}{k} \left(\frac{1}{N}\right)^k \left(\frac{N-1}{N}\right)^{n-k} \frac{k}{nb}$$

$$= \frac{1}{nb} \sum_{k=1}^{n} \binom{n}{k} \left(\frac{1}{N}\right)^k \left(\frac{N-1}{N}\right)^{n-k} k =$$

$$= \frac{1}{nb} \mathbb{E}[z_i] = \frac{1}{nb} \frac{n}{N} = \frac{1}{Nb}$$

Observe that a sample from $S$ has the same distribution as a sample from the dataset itself, which is bound by $\sigma^2$ according to assumption 5 in Section 3.1.

Overall we get

$$\mathbb{E}_S\left[\mathrm{V}_{\widetilde{B}_l}\left(\nabla f_{\widetilde{B}_l}|S\right)\right] = \frac{1}{b}\mathbb{E}_S[\mathrm{V}_i\left(S_i|s\right)] \leq \frac{1}{b}\left(\sigma^2 + \frac{1}{n}h_D \frac{\sigma^2}{b}\right)$$

And the variance reduction of Theorem 1 is achieved by plugging in $(i)$ and $(ii)$

## B.2  THEOREM 2

**Theorem 3.** *Suppose that $F(\mathbf{x})$ are smooth and $\mu$-strongly convex functions. Let $T = \mathcal{T}nb$ be the total number of examples seen during training, where $\mathcal{T} \geq 1$ is the number of buffers iterated. Choose the learning rate to be $\eta_t = \frac{6}{bn\mu(t+a)}$ where $a \geq \max\left\{\frac{8LG+24L^2+28L_HG}{\mu^2}, \frac{24L}{\mu}\right\}$. Then, Corgi$^2$, has the following convergence rate in the online stage for any choice of $\mathbf{x}_0$,*

$$\mathbb{E}[F\left(\bar{\mathbf{x}}_{\mathcal{T}}\right) - F(\mathbf{x}^*)] \leq (1-\alpha)h_D'\sigma^2 \frac{1}{T} + \beta\frac{1}{T^2} + \gamma\frac{(Nb)^3}{T^3}, \tag{10}$$

*where $\bar{\mathbf{x}}_{\mathcal{T}} = \frac{\sum_t (t+a)^3 \mathbf{x}_t}{\sum_t (t+a)^3}$, and*

$$\alpha := \frac{n-1}{N-1}, \beta := \alpha^2 + (1-\alpha)^2(b-1)^2, \gamma := \frac{n^3}{N^3}.$$

### B.2.1  PROOF

This theorem corresponds very closely to Theorem 1 in (Xu et al., 2022a). In fact, our proof is not a complete derivation of the convergence rate, but rather an application of the variance reduction obtained in Theorem 1 to the existing convergence rate derived for CorgiPile.

Since the online phase of Corgi$^2$ is identical to CorgiPile, most of the logic used in deriving the convergence rate for CorgiPile should be applicable for Corgi$^2$. However, in CorgiPile the dataset itself is non stochastic, while Corgi$^2$ generates it in the offline phase, introducing new randomness. The CorgiPile convergence rate is:

$$\mathbb{E}_{\text{CogiPile}}[F\left(\bar{\mathbf{x}}_{\mathcal{T}}\right) - F(\mathbf{x}^*)] \leq (1-\alpha)h_D\sigma^2 \frac{1}{T} + \beta\frac{1}{T^2} + \gamma\frac{(Nb)^3}{T^3} \tag{11}$$

and our modification can be seen as taking the expected value over the offline randomness:

$$\mathbb{E}_{\text{OfflineCorgiShuffle}}[\mathbb{E}_{\text{CogiPile}}[F\left(\bar{\mathbf{x}}_{\mathcal{T}}\right) - F(\mathbf{x}^*)]] \leq? \tag{12}$$

The following proof does not provide a comprehensive reconstruction of the one for CorgiPile convergence rate(Xu et al., 2022b), because The majority of steps in that proof remain unaffected when encapsulated within a new expectation expression. Consequently, the subsequent section of this proof will refer directly to sections in the CorgiPile convergence rate proof without providing complete statements here.

An essential observation to note is that assumptions 1-4 in Section 3.1 impose *upper bounds* on properties of *all* individual or pairs of samples from the dataset. Given that OfflineCorgiShuffle (Algorithm 2) outputs a subset (with repetitions) of the origianl dataset, these assumptions are ensured to remain valid. For this reason, any step in the CorgiPile proof which replaces an expression with $L$, $G$ or $H$ works as-is for the Corgi$^2$ proof.

The CorgiPile proof begins by taking a known upper bound on $\mathbb{E}_{\text{CorgiPile}}\big[||X_0^{t+1} - X^*||^2\big]$, and develops it untilthe following inequality is reached:

$$\underbrace{\eta_t bn\left(F(X_0^t) - F(X^*)\right)}_{I} \leq \underbrace{\left(1 - \frac{1}{2}\eta_t bn\mu\right)||X_0^t - X^*||^2 - \mathbb{E}\big[||X_0^{t+1} - X^*||^2\big]}_{II} + \underbrace{C_2\eta_t^2 nb\frac{N-n}{N-1}h_D\sigma^2}_{III}$$

$$+ \underbrace{C3\eta_t^3 bn\left[\left(\frac{N-n}{N-1}\right)^2(b-1)^2 + \left(\frac{N-1}{N-1}\right)^2\right] + C_4\eta_t^4 b^4 n^4}_{IV}$$

Note that the notation here differs slightly from the one that can be found in the CorgiPile paper as we use $t$ to denote the round number instead of $s$. Additionally, we'll use $\mathbb{E}_{S,\widetilde{B}}[]$ to express taking an expected value over the randomness of OfflineCorgiShuffle.

Taking the expectation over OfflineCorgiShuffle randomness on both sides of this inequality has the following effects:

- I
$$\mathbb{E}_{S,\widetilde{B}}\big[\eta_t bn\left(F(X_0^t) - F(X^*)\right)\big] = \eta_t bn\mathbb{E}_{S,\widetilde{B}}\big[\left(F(X_0^t) - F(X^*)\right)\big]$$

- II
$$\mathbb{E}_{S,\widetilde{B}}\left[\left(1 - \frac{1}{2}\eta_t bn\mu\right)||X_0^t - X^*||^2 - \mathbb{E}\big[||X_0^{t+1} - X^*||^2\big]\right]$$
$$= \left(1 - \frac{1}{2}\eta_t bn\mu\right)\mathbb{E}_{S,\widetilde{B}}\big[||X_0^t - X^*||^2\big] - \mathbb{E}_{S,\widetilde{B}}\big[\mathbb{E}\big[||X_0^{t+1} - X^*||^2\big]\big]$$

- III - see discussion below

- IV
$$\mathbb{E}_{S,\widetilde{B}}\left[C3\eta_t^3 bn\left[\left(\frac{N-n}{N-1}\right)^2(b-1)^2 + \left(\frac{N-1}{N-1}\right)^2\right] + C_4\eta_t^4 b^4 n^4\right]$$
$$= C3\eta_t^3 bn\left[\left(\frac{N-n}{N-1}\right)^2(b-1)^2 + \left(\frac{N-1}{N-1}\right)^2\right] + C_4\eta_t^4 b^4 n^4$$

As for III, both $h_D$ and $\sigma^2$ cannot be treated as constants in the context of $\mathbb{E}_{S,\widetilde{B}}[\cdot]$. $\sigma^2$ is affected by OfflineCorgiShuffle because the relating dataset is a subset (potentially with repetitions) of the original, and thus may have a different variance. $h_D$ is affected because the new blocks are not guaranteed to have the same blockwise variance (in fact, changing $h_D$ is the primary gain of OfflineCorgiShuffle).

Instead of attempting to to compute the expectation of this component, we will inspect an earlier part of this proof, at the point where the component was first derived - specifically, the calculation of $I_4$, at the inequality in equation (10):

$$I_4 = 2\eta_t^2 \mathbb{E}\left[||\sum_{k=1}^{bn} \nabla f_{\Psi_t(k)} - \mathbb{E}\left[\sum_{k=1}^{bn} \nabla f_{\Psi_t(k)}\right]||^2\right] = \frac{n(N-n)}{N-1}\mathbb{E}_j\left[||\nabla f_{B_j}(X_0^t) - b\nabla F(X_0^t)||^2\right]$$

$$\leq s\eta_s^2 \frac{nb(N-n)}{N-1}h_D\sigma^2$$

where the inequality is due to assumption 5 in Section 3.1. Due to Theorem 1 we have:

$$\mathbb{E}_{S,\widetilde{B}}\left[\frac{n(N-n)}{N-1}\mathbb{E}_j\left[||\nabla f_{B_j}(X_0^t) - b\nabla F(X_0^t)||^2\right]\right] = \frac{n(N-n)}{N-1}\mathbb{E}_{S,\widetilde{B}}\left[\mathbb{E}_j\left[||\nabla f_{B_j}(X_0^t) - b\nabla F(X_0^t)||^2\right]\right]$$

$$\leq \frac{n(N-n)}{N-1}h_D'\frac{\sigma^2}{b}$$

Thus we may substitute $\mathbb{E}_{S,\widetilde{B}}\left[C_2\eta_s^2 nb\frac{N-n}{N-1}h_D\sigma^2\right]$ with $C_2\eta_s^2 nb\frac{N-n}{N-1}h_D'\sigma^2$. All in all we obtain:

$$\eta_s bn\mathbb{E}_{S,\widetilde{B}}\left[(F(X_0^t) - F(X^*))\right] \leq \left(1 - \frac{1}{2}\eta_s bn\mu\right)\mathbb{E}_{S,\widetilde{B}}\left[||X_0^t - X^*||^2\right] - \mathbb{E}_{S,\widetilde{B}}\left[\mathbb{E}\left[||X_0^{t+1} - X^*||^2\right]\right]$$

$$+ C_2\eta_t^2 nb\frac{N-n}{N-1}h_D'\sigma^2 + C3\eta_t^3 bn\left[\left(\frac{N-n}{N-1}\right)^2(b-1)^2 + \left(\frac{N-1}{N-1}\right)^2\right]$$

$$+ C_4\eta_t^4 b^4 n^4$$

The CorgiPile proof proceeds by applying lemma 3 (from their article) where series $a$ is $\{(F(X_0^t) - F(X^*))\}_t$ and series $b$ is $\{||X_0^t - X^*||^2\}$. In our case, $\mathbb{E}_{S,\widetilde{B}}[(F(X_0^t) - F(X^*))]$ and $\mathbb{E}_{S,\widetilde{B}}\left[||X_0^t - X^*||^2\right]$ can be used as seamless replacements.

From that point forward no additional modifications are required, and we arrive at the convergence rate described in Theorem 2.

## C EXPERIMENTAL DETAILS

We implemented Corgi$^2$ shuffler within the PyTorch framework. Our code will be available online. The indexes of the dataset are allocated to blocks, which are then shuffled in either CorgiPile, Corgi$^2$ or full shuffle before the training.

For CIFAR-100, we trained ResNet-18 for 200 epochs with a batch size of 256, learning rate 0.1, momentum 0.9, weight decay $5e-4$, and a Cosine Annealing LR scheduler. We used standard data augmentations - random crops, horizontal flips, rotations, and normalization (with the standard mean and std for CIFAR-100).

For ImageNet, we trained ResNet-50 for 100 epochs with a batch size 2048, learning rate 0.1, momentum 0.9 weight decay $1e-4$, and a Cosine Annealing LR scheduler. We used the PyTorch AutoAugment functionality followed by random horizontal flip and normalization (with the standard mean and std for ImageNet) for data augmentation.

We changed the parameters $n$ and $b$ to fit the target buffer ratio for each experiment, but maintained the values when comparing CorgiPile to Corgi$^2$ on the same buffer ratio.

### C.1 RESULTS DISCUSSION

Overall the experimental results affirm that Corgi$^2$ significantly outperforms CorgiPile. However, it may appear surprising that it did considerably better on ImageNet - where it achieved parity with full shuffle - than on CIFAR-100, where performance was degraded compared to full shuffle.

We believe this occurred as a result of using artificially small buffer ratios on a dataset that wasn't large to begin with. Up to this point we've never considered the specific task a learning model is

trying to accomplish, and have thus focused on the variance between blocks as a key metric. However, CIFAR100 is a classifier. It is well known ((Zhang et al., 2021)) that a dataset with an imbalanced weighting among classes (i.e the data is not equally distributed among classes) imposes additional challenges on the training process. By limiting the buffer size to 0.2% on CIFAR100, one ends up with 100 examples per buffer (out of a total of 50000 in the train set) - which would lead to a high variance of the weight balancing among classes, compounding on top of the usual increase in variance that CorgiPile and Corgi$^2$ impose. While we have not quantified this in either a theoretical or experimental manner, it is reasonable to expect that this would slow down the convergence rate, which is the phenomena we observe in our results.

While undesirable, this scenario does not appear to be very realistic - in most non synthetic cases, the number of examples that can fit on the memory of a single worker would be considerably larger than the number of classes.

