# OpenReview forum: "Corgi$^2$: A Hybrid Offline-Online Approach To Storage-Aware Data Shuffling For SGD"
_ICLR.cc/2024/Conference — Submitted to ICLR 2024_

### Official Review · Reviewer_xhMF · 2023-10-26

**Soundness:** 3 good
**Presentation:** 3 good
**Contribution:** 2 fair
**Rating:** 6
**Confidence:** 3

**Summary:**

This paper addresses the problem of accessing large datasets in an efficient manner without compromising on the accuracy of SGD. In particular, for datasets stored on the cloud in homogeneous shards, the authors try to address the performance degradation of CorgiPile, which is an online shuffling algorithm. The authors design an extra offline shuffling procedure, which can be implemented without requiring extra storage, that diversifies the examples seen during the training. Depending on block sizes $b$, the provable speed-up relative to CorgiPile can be much larger than 1. The authors then empirically study the test accuracy accuracyon standard benchmarks of training ResNet on CIFAR100 and ImageNet, and, in addition, the test loss when training on a proprietary dataset.

**Strengths:**

1. The problem of efficient data partitioning and loading has a high potential. If we can achieve an improvement on these steps, it might affect all of large-scale deep learning.
2. The authors gave a theoretical justification for why their method can be better and quantified the speed-up.
3. There seems to be almost no downside (other than some extra work before training) to the proposed shuffling procedure.
4. The method is tested empirically on real-world problems.

**Weaknesses:**

1. The assumptions for the theoretical analysis do not look reasonable. While SGD without replacement has been studied with various assumptions, state-of-the-art results (convex and nonconvex) rely only on smoothness and bounded variance. This work, on the other hand, requires bounded gradients and Lipschitz Hessians. It seems that the reason is that the authors used the suboptimal theory of Xu et al. (2022a).
2. The experiments are a bit limited (only vision models) and the authors report only a single random seed.
3. I do not see any empirical evaluation of how much time (relative to the whole training time) we can save by using Corgi2 instead of shuffling the data once before the training.

## Minor
The abstract ends with ". paragraph."
"and Nguyen et al. (2022), where" -> "and (Nguyen et al., 2022), where"
"illustrate a a qualitative" -> "illustrate a qualitative"
I found Figure 2 to be rather confusing than helping, I was particularly confused as to where Sample 7 came from and what black color means.
"to minimize an average of functions" -> "to minimize the average of functions"
"Sample an example $f_i$ from the dataset, where $i$ is selected uniformly at random." Since you're talking about running SGD in epochs, $i$ should not be selected uniformly but rather it should be sampled without replacement until we process the full dataset. Otherwise, epochs shouldn't be mentioned here.
"left hand side" -> "left-hand side"
I was completely unfamiliar with the name "Bienaymé’s identity" for the variance of the sum, I'd suggest the authors clarify its meaning in the text or simple write the identity there
The identity (ii) on page 6 is missing a "." at the end
Theorem 2, "$F(x)$ are" -> "$F(x)$ is"
"While scenarios where this modification is useful may very well exists" I did not understand the meaning of this sentence
"in section 2" -> "in Section 2"
Please fix the references, for example, "Pablo A Parrilo" -> "Pablo A. Parrilo", "sgd" -> "SGD", add the name in "S Kocherlakota", etc.
The punctuation in the appendix is currently missing

**Questions:**

1. Why you didn't run CorgiPile on the proprietary dataset?
2. I might be mistaken, but it appears to me that the requirement to sample with replacement to provide a theoretical guarantee can be actually lifted. In particular, using Lemma 1 in (Mishchenko et al, "Random Reshuffling: Simple Analysis with Vast Improvements"), you can directly bound the variance of the sum without replacement. This should resolve the issues mentioned in Section 5.
3. Can the authors measure the empirical speed-up achieved by using Corgi2 instead of full shuffle?

---

### Official Review · Reviewer_FYCm · 2023-10-29

**Soundness:** 2 fair
**Presentation:** 2 fair
**Contribution:** 1 poor
**Rating:** 3
**Confidence:** 4

**Summary:**

This work focuses on how to get rid of the full shuffle on the entire dataset before training. To be specific, this work proposes $Corgi^2$, which is based on a previous partial shuffling method Corgipile. Compared with Corgipile, $Corgi^2$ applies the partial shuffling both before training (offline) and during the training (offline) to increase the randomness of data access.

**Strengths:**

- Detailed theoretical analysis upon the convergence.
- Has experiments.

**Weaknesses:**

(1) The viewpoint that full shuffle on the entire dataset before training is costly is a bit doubtful for me. In practice, one may train on the same dataset for multiple times to tune the hyper-parameters, and thus the full shuffle time is less important. Furthermore, the proposed method simply applies the Corgipile method twice and lacks novelty.

(2) The analyzed problem in Section 3 is similar to the one in the following paper. Please cite and discuss appropriately.
- Ramezani, Morteza, et al. "Gcn meets gpu: Decoupling “when to sample” from “how to sample”." Advances in Neural Information Processing Systems 33 (2020): 18482-18492.

(3) Only convergence w.r.t. training epochs is reported in the experiments. The running time, including the time of offline full shuffle, the time of offline corgi shuffle (Alg 2), the training time of each iteration/epoch for each compared method, and the convergence w.r.t. training time, should be reported accordingly. Additionally, the results of Corgipile is not reported in Figure 5.

(4) Minor comments:
- A dangling “paragraph” at the end of abstract.
- The caption of Figure 3 (right) is “ResNet18 on ImageNet”. Shoud it be “ResNet18 on CIFAR-100”?
- A conclusion section seems missing.

**Questions:**

See weaknesses.

---

### Official Review · Reviewer_ZuF5 · 2023-10-31

**Soundness:** 2 fair
**Presentation:** 3 good
**Contribution:** 2 fair
**Rating:** 3
**Confidence:** 4

**Summary:**

This paper proposes an algorithm named Corgi$^2$, a variant of SGD tailored for large datasets stored in multiple distributed storage blocks. The algorithm is an extension of an existing algorithm CorgiPile proposed for in the same setup. Corgi$^2$ adds an offline shuffling (called OfflineCorgiShuffle) before executing CorgiPile. The offline shuffling mixes data points across different storage blocks, hence making the distribution of data points stored in each block more similar to one another. This reduction of inter-block variability leads to better theoretical guarantees in the convergence speed of Corgi$^2$ compared to CorgiPile.

**Strengths:**

S1. The paper studies an algorithm relevant to practical scenarios and proposes a scheme that improves upon an existing algorithm CorgiPile.

S2. The paper is well-written and is easy to follow.

**Weaknesses:**

W1. I think the main weakness of this paper is that its contributions are a bit incremental. The key contribution is to add an offline step to mix between different data blocks, and the analysis of the reduced block variance. After that, everything else is a straightforward application of the existing bound by CorgiPile, with the block variance term replaced with a smaller one. For this reason, I think the overall technical novelty is limited.

W2. I also question if the proposed offline shuffling step is realistic. First, the with-replacement version of Algorithm 2 requires creating *new* storage blocks to store the new blocks, doubling the storage requirement. This is a serious limitation in practice, as the algorithm is more pertinent to the setting where the dataset is too large so that it has to be stored in a distributed manner. I reckon that developing an analysis for the without-replacement and in-place version of Algorithm 2 must be carried out for this algorithm to have any practical impact. Even in the case where in-place implementation can be adopted, I still question if it actually makes sense to mix between different storage blocks. This is directly against the philosophy of in-database machine learning which was the original motivation for the development of CorgiPile.

W3. The "baseline" SGD is not clear throughout the paper, as the paper switches between random access SGD (with-replacement SGD, where index is chosen uniformly at random) and full shuffle SGD (without-replacement SGD, where index is shuffled and then accessed sequentially). Figure 1 compares the proposed algorithm against full shuffle, Section 2 describes random access SGD and Section 3.2.1 mentions it as the baseline, but experimental evaluation is done against full shuffle SGD.

W4. The paper fails to cite the recent development in the analysis of shuffling SGD, and I think this is a bad omission. Over the last several years, it has been shown that for shuffling SGD variants such as Random Reshuffling (RR) and Shuffle Once (SO) converge **faster** than random access SGD [A, B, C, and more] in certain cases. For example, when component functions $f_i$ are smooth and the average $F$ is strongly convex (as assumed in this paper), both RR and SO converge at the rate of $\tilde O(\frac{1}{nK^2})$ when the number of epochs $K$ is large enough. One can notice that this is indeed faster than $O(\frac{1}{T})$ rate of random access SGD, by realizing that the number of iterations $T$ satisfies $T = nK$. Given this set of results highlighting the difference between full shuffle SGD vs random access SGD, I believe the actual theoretical baseline for Corgi$^2$ should be the full shuffle variant of SGD.

[A] Random Reshuffling: Simple Analysis with Vast Improvements

[B] SGD with shuffling: optimal rates without component convexity and large epoch requirements

[C] Tighter Lower Bounds for Shuffling SGD: Random Permutations and Beyond

**Questions:**

Q. I am not fully convinced by the proof sketch in Section 3.2. In particular, I am not sure why we get the $-\frac{1}{nb}\cdot \frac{\sigma^2}{b}$ part in the RHS of the last inequality. I looked up the full proof in Appendix B but the proof there has $+\frac{1}{nb}\cdot h_D \frac{\sigma^2}{b}$, a slightly different term with the *opposite* sign.

---

### Official Review · Reviewer_neFw · 2023-11-05

**Soundness:** 2 fair
**Presentation:** 3 good
**Contribution:** 3 good
**Rating:** 5
**Confidence:** 3

**Summary:**

This paper proposes Corgi$^2$, an approach of mitigating the limitation of previously proposed algorithm CorgiPile by adding another offline step which incurs a small overhead. The authors give detailed theoretical analysis, which elucidates the conditions under which Corgi$^2$ converges and offers insights into the trade-offs between data access efficiency and optimization performance.

**Strengths:**

* Data shuffling is a timely and important topic in model training. And the motivation of adding an offline step, with small overhead, to address the limitation of CorgiPile sounds promising.
* The theoretical analysis is detailed and insightful.

**Weaknesses:**

* In the theoretical analysis, usually bounded gradient and bounded variance are not assumed at the same time. Is it critical for Corgi$^2$ to converge based on both conditions?
* The experimental section lacks comparison of wall-clock time convergence. Test accuracy or training loss w.r.t time as X-axis is missing. This is critical since the main benefit of the proposed algorithm relies on the fact that the offline step is not too expensive.
* The paper lacks evidence that the additional $2m/b$ queries of Corgi$^2$ compared to CorgiPile does not introduce overhead. Please consider adding some real-world measuring.

**Questions:**

I'd be happy to increase my rating if the weaknesses part can be properly addressed.

---

### Meta-Review · Area_Chair_zAhh · 2023-12-06

**Metareview:**

The authors did not provide the rebuttal to address the reviewers’ concerns. Therefore, the paper cannot be accepted with the current reviews.

**Justification For Why Not Higher Score:**

No response from the authors.

**Justification For Why Not Lower Score:**

N/A

---

### Decision · Program_Chairs · 2024-01-16

Reject